# Utilizing 3D Arterial Spin Labeling to Identify Cerebrovascular Leak and Glymphatic Obstruction in Neurodegenerative Disease

**DOI:** 10.3390/diagnostics11101888

**Published:** 2021-10-13

**Authors:** Charles R. Joseph

**Affiliations:** Department of Internal Medicine, Liberty University College of Osteopathic Medicine, Lynchburg, VA 24502, USA; crjoseph@liberty.edu

**Keywords:** glymphatic flow, blood brain barrier, Alzheimer disease, neurodegenerative diseases, 3D ASL MRI, dynamic contrast Imaging

## Abstract

New approaches are required to successfully intervene therapeutically in neurodegenerative diseases. Addressing the earliest phases of disease, blood brain barrier (BBB) leak before the accumulation of misfolded proteins has significant potential for success. To do so, however, a reliable, noninvasive and economical test is required. There are two potential methods of identifying the BBB fluid leak that results in the accumulation of normally excluded substances which alter neuropil metabolism, protein synthesis and degradation with buildup of misfolded toxic proteins. The pros and cons of dynamic contrast imaging (DCI or DCE) and 3D TGSE PASL are discussed as potential early identifying methods. The results of prior publications of the 3D ASL technique and an overview of the associated physiologic challenges are discussed. Either method may serve well as reliable physiologic markers as novel therapeutic interventions directed at the vasculopathy of early neurodegenerative disease are developed. They may serve well in addressing other neurologic diseases associated with either vascular leak and/or reduced glymphatic flow.

## 1. Introduction

In the wake of disappointing treatment trials of Alzheimer modifying drugs, a fresh therapeutic approach to this and other neurodegenerative diseases is necessary. Approaching the initial disease phase, the pathological leak of the blood brain barrier and consequent obstruction of glymphatic flow is worthy of therapeutic investigation. To do so requires identification of these very early pathophysiologic changes developing before the accumulation of misfolded proteins and significant cognitive decline has developed. [1,2]. Looking at sporadic AD in stages, the initial insult is the blood brain barrier (BBB) leak, triggered by endothelial and pericyte damage resulting from circulating low level upregulated pro-inflammatory factors via several potential sources [1,3]. Senescence, morbid obesity, chronic infection, autoimmune disease, diabetes mellitus, hypertension and head injury are the most common factors [3,4,5]. Similar to a computer hacker, the attacks are relentless over one’s lifetime and once the defenses are overwhelmed, either because of sheer volume of pro-inflammatory factors or reduced effectiveness of the CNS immune response (e.g., microglia and astrocyte responses), the BBB is breached [6,7]. The circulating inflammatory factors cause endothelial cell up regulation of Matrix Metallo Protease Enzyme-9 (MMPE-9), expressed on the cell surface triggering release of reactive oxygen species ROS from circulating and inherent macrophages and immune cells. Damage to the tight junctions, endothelium and pericytes results [8,9]. The consequence is a leak into the interstitium of normally excluded substances, so called damage associated molecular patterns (DAMPS) and pathogen associated molecular patterns (PAMPS) which alter the normal metabolic machinery resulting in distorted protein synthesis and degradation [10,11,12]. Additionally, both misfolded proteins and toxins overwhelm the innate immune cells (microglia and macrophages) normally clearing them, either by degradation or by outflow from the venous or glymphatic systems (Figure 1a,b) [13,14,15,16]. As a consequence of the pericyte damage that normally anchors the aquaporin 4 water channels to astrocyte end feet via expressed laminin and dystroglycan proteins, the glymphatic drainage system effectively shuts down as the untethered AQ 4 channels return to the astrocyte soma. The glymphatic system is a major pathway of metabolite clearance from the brain parenchyma (Figure 1 and Figure 2) [17,18].

The entrance of normally restricted substances into the brain leads to the second stage of disease, the miscleavage of APP probably through intermediate steps and subsequent misfolding of beta amyloid fragments 1–42 followed by crosslinking and extracellular precipitation. Its presence, in time, facilitates production of hyperphosphorylated Tau (hpTau) [19,20,21]. The accumulation of the latter substance is the 3rd and terminal phase of the disease process development [22,23,24]. Because of the deadly nature of hpTau and its propensity to spread transsynaptically as well as by exostosis in a Prion-like fashion, its appearance is the irretrievable end of the road [24]. An apt analogy is considering a pristine lake filled with fish and wildlife fed by a clear stream with waste products draining via an outflow stream. If the water source from the clear stream becomes polluted, over time the lake does as well and the fish begin to die, thus adding to the pollution as well as plugging up the outflow. If one were to simply remove the dead fish but not stop the polluted water inflow, the effect on lake pollution would be minimal, if any, and more wildlife would perish. So, too, removing beta amyloid and hpTau is similar to removing the dead fish and thus, not surprisingly, has little effect on altering disease progression or recovery. It is in the early phase of the disease that repairing the BBB leak must be addressed before the overwhelming effect of toxin induced metabolic disarray and accumulating misfolded proteins develop [25,26]. To do so requires a reliable method of identifying the altered BBB/glymphatic flow physiology. Testing must identify evidence of fluid leak into the parenchyma and/or reduced outflow via the glymphatic system.

An orderly approach to identifying early sporadic AD includes surveillance of high-risk groups such as those with poorly controlled Diabetes mellitus, hypertension, head injury, morbid obesity, glioblastoma, advanced age and those with chronic inflammatory/infectious processes or identified high risk self-limited infections [27,28,29]. A reliable noninvasive, simple, economic and reliable screening test is desirable to that end.

At this point, it is useful to consider far less common familial AD (accounting for about 5% of all AD), the most common being APOE4 [30,31,32]. If we view these pathways of disease development as being from the inside out as opposed to the outside in (sporadic disease), one must modify the lake analogy to the pollution source being within the lake itself. This may have profound implications as to treatment strategies since stopping the BBB leak would theoretically be less efficacious in the familial group due to the continuous and inherent production of beta amyloid and its negative impact on vascular integrity and the BBB [33,34]. Having a reliable noninvasive screening test for identifying a BBB leak and glymphatic dysfunction is still necessary even though treatment strategy may differ from sporadic disease. That said, in either case, following the progress of treatment trials’ success requires demonstrating both improvement in glymphatic flow and reduced BBB leak coupled with stable or improved cognitive testing.

Investigating differences in perfusion in neurodegenerative diseases is one approach, either by DCE or ASL (Table 1).

The direct method, high resolution dynamic contrast enhancement/imaging (DCI or DCE), is to image the presence of fluid leaking past the BBB into the interstitium, either by direct or indirect means. DCE is a post gadolinium contrast infusion technique [35,36,37,38,39,40,41]. It utilizes fast image acquisition to sequentially measure the entrance of contrast into the field of view and it can determine a variety of physiologic parameters related to blood flow and perfusion [42]. Barnes et al. utilized high resolution DCE to image BBB leak of contrast into the hippocampus parenchyma [43]. They compared normal subjects of varying ages with mildly cognitive impaired subjects (MCI) and those with AD. The study utilized high resolution T2-weighted images through the hippocampi and baseline coronal T1-weighted maps acquired using a T1-weighted 3D spoiled gradient echo (SPGR) pulse sequence [2,43]. In addition, coronal DCE-MRI scans were obtained using a T1-weighted 3D SPGR pulse sequence repeated for 16 min with 15.4 s temporal resolution per image. Voxel size was 0.625 × 0.625 × 5 mm. The results demonstrated BBB leak in older age normal subjects in MCI compared with controls and in the AD subjects also compared to controls. Via the technique, the K_transfer_ of contrast into the parenchyma was calculated [44]. This is a landmark study demonstrating via imaging clear evidence of the BBB disruption in the preclinical and clinical phases of the disease process [2,43]. The technique itself may require long scan times too difficult to adopt in clinical practice. Long sequence duration also invites movement artifact. This method requires infusion of gadolinium contrast, which is both costly and not without risk.

## 2. Perfusion ASL

One ASL approach measures perfusion inflow at peak, which has been shown to be reduced in AD. Measurements are hampered by differences in perfusion between white and gray matter and thus sampling errors. This makes for difficulties in intersubject comparison related to the relative ratio of gray and white matter within the area of interest sampled. Further, only one time to inversion is obtained at the theoretic peak of perfusion. Correction algorithms are available but require further validation [44]. Our alternative 3D TGSE PASL technique measuring signal at post perfusion intervals is discussed below. The two current methods for identifying BBB leak are summarized in Table 1.

## 3. 3D-ASL Method

Another method is to consider the effect of the BBB breach and its early consequence of reduced glymphatic flow. Arterial spin labeling allows for noninvasive identification of perfusion in and venous flow out of the brain [45,46]. Utilizing arterial spin labeling and timing the data collection to the post arterial inflow and capillary phases, theoretically, would allow imaging of retained labeled protons primarily within the interstitium and, to a lesser extent, within the venous and glymphatic channels requiring progressively longer inversion intervals from labeling than is used in inflow perfusion. From this, a rate of residual clearance can be determined [46]. A reduction in the outflow rate, if present, can be accounted for by both leak in and sequestration of labeled protons within the interstitium and diminished glymphatic outflow [26,46,47]. There are two main obstacles to this approach. The first is knowing the T1 of the environments in which labeled protons will migrate post bolus and the differential flow velocities of arterial, venous and glymphatics [46,47]. As to the latter concern, choosing time to inversion late in the blood transit cycle reduces contamination from labeled arterial and the majority of venous flow (<2 s/bolus), leaving signal primarily from residual interstitial and glymphatic fluid labeled protons [48,49]. The T1 at 3T of free fluid is considerably longer (3800 ms) than gray matter (1100–1700 ms), white matter (800–850 ms) or blood (1650 ms), a fact to one’s advantage [49,50]. The flip side, however, is avoiding the ventricular and subarachnoid spaces in the region of interest (ROI) analysis as this can introduce considerable error. The other major problem with long inversion times is reduced signal to noise (S/N) [51].

The transit time of the labeled bolus is about 1.8 s, with contributions to the remaining signal at longer post labeling intervals from mainly labeled fluid in extra-capillary proton environments, which has been measured by other techniques [45,48]. By intentionally choosing longer post labeling delay times (PLD or TI), signal from retained leaked protons within interstitial fluid and the slower glymphatic flow is maximized. By measuring clearance of fluid outflow, a glymphatic flow rate reduction compared with normal controls is indirect evidence of both a labeled proton leak into the interstitial spaces and their reduced outflow via damaged glymphatics [46]. By choosing longer inversion times (in our study, TI’s of 2800–4000 ms at 200 ms intervals past the T1 of blood), the signal contribution of gray and white matter and intravascular elements is minimized and residual labeled brain water signal with the longest T1 is maximized [50,52]. Because the S/N in these longer inversion times is low, a single determination at any specified time point would result in too much variability for quantitative interpretation [45]. By measuring several sequential inversion times, the results can be graphed with the slope of the line indicating the rate of signal loss or clearance from natural signal decay or glymphatic outflow. Linear analysis is possible if the delay times correspond to more linear aspects of the signal decay curve, which can be determined by solving the Block equation for each tissue and fluid component environment within the neuropil [46]. Doing so demonstrated that linear analysis in these late delayed signal acquisitions showed high correlation (95%) with the signal decay curve in all relevant tissues and liquid proton environments [46]. Thus, residual signal is emitted largely from labeled free fluid protons trapped within the interstitium due to impaired glymphatic outflow [26].

For wide adoption of ASL MRI into clinical use, several basic requirements must be met. The first is hardware with access to a high field strength magnet 3T or higher and adequate gradients to allow for fast signal acquisition (EPI) sequences [53]. The sequence protocol has to be both time and cost efficient. Rapid filling of K-space, allowing for side-by-side analysis of homologous and different brain regions near simultaneously, avoids signal transit time disparities within the field of view. Using 3D acquisition with near instantaneous signal acquisition of the whole brain permits sampling and direct comparison of signal within individual lobes. Additionally, this noninvasive technology eliminates the cost and risk of contrast agents when multiple studies are required over time. Finally, reproducibility is the final concern and, in the pilot study, did not pose a problem. Siemens 3D TGSE PASL (turbo gradient spin echo pulsed arterial spin labeling) sequence fulfills the above criterion but at the cost of reduced S/N. The latter issue precludes simple qualitative analysis and requires a quantitative approach. First, to compensate for low S/N, sampling encompassed a large region of interest (ROI) in the chosen anatomic site with careful exclusion/minimization of ventricular or subarachnoid spaces, since residual signal within the parenchyma is of interest only. Second, acquiring multiple sequential data points (seven in the pilot study) to generate the linear analysis reduces error [46].

The sequence availability is dependent on the manufacturer and installed gradients. The protocol available to us on the Siemens 3T Skyra magnet was 3D TGSE PASL. To maximize signal, a 20 channel receiver head coil was used [45]. Addressing the multiple requirements listed above were several features of this standard available sequence, which we will discuss. 3D ASL has been extensively reviewed elsewhere, with the primary recommendation for implementation being dictated by the process of labeling [53]. Considered the two most efficient are pseudo-continuous arterial spin labeling (pCASL), with the longest duration of the labeling pulse, followed by pulsed arterial spin labeling (PASL) [53]. A critical feature is determining the precise time to inversion for which pCASL has advantage but is compensated for in PASL by adding flow-sensitive alternating inversion recovery (FAIR) [54,55].

Beginning with the labeling portion of the sequence, FAIR allows for acceptable perfusion contrast by effectively reducing the travel distance of tagged protons [54,55]. Likewise, quantitative imaging of perfusion using a single subtraction (Q2TIPS) minimizes the effect of varying arrival times of the labeled bolus to the parenchyma, reducing another variable [56]. The number of label-control pairs is also a consideration as the more averaged the better the resolution but with added time of acquisition penalty and thus artifact. The sequences designed by the vendor have optimized the number of pairs required. Four pairs were averaged in the Siemens 3D sequence used in the pilot study [46].

3D ASL for this purpose has several advantages over 2D ASL imaging, including more signal acquired [57]. For example, the ability to acquire the whole head, filling K-space in a single acquisition and allowing for comparison of various parts of the brain simultaneously with reliability for the time of data acquisition. That would not be the case with a 2D acquisition sequence where each slice would be acquired at different acquisition times [57]. In addition, the combination of an EPI factor of 21 with 12 segments (multi-shot EPI) and turbo factor of 18 greatly reduces acquisition time (350 ms total) for each sequence [58]. This fulfills another requirement for comparing the rate of clearance among multiple brain regions near simultaneously. The advantage of a high turbo and EPI factor translates into reduced exam time. Our protocol takes about 15 min including 7 sequential ASL sequences with inversion times starting at 2800 ms increasing by 200 ms to 4000 ms along with a reference fluid attenuated inversion recovery (FLAIR) axial sequence [46]. No exogenous contrast agent is required.

## 4. Results

In our initial pilot study, this approach demonstrated statistically significant slowing of the glymphatic fluid clearance rate per second in a small sample of Alzheimer disease subjects compared with age matched controls (Figure 3) [46]. The sampled area included bilateral temporal, parietal and frontal lobes. All but the dominant temporal lobe showed difference and showed significance to the level *p* < 0.001 when comparing the AD subjects with age matched controls. The dominant temporal lobe in the comparison showed *p* = 0.068 [46]. The latter may be by chance or another unrecognized mechanism which may be sorted out with a larger study (in progress).

## 5. Discussion

Utilizing 3D ASL MRI as an indirect measure of vascular leak and reduced glymphatic flow or DCE as a direct measure of vascular leak in neurodegenerative diseases will address these early issues. Treatments that arrest the slow vascular leak may prevent the disease cascade from developing, especially in the sporadic forms where accumulating misfolded proteins is a downstream effect, and a component of familial AD [25,26,35,59]. To assess outcomes of directed therapeutic trials, identification of disease in the pre or early clinical state becomes paramount. We are currently conducting a larger study to further validate the 3D ASL technique. Included in the latest study are patients with mild cognitive impairment. Should the latter group demonstrate focal regional changes consistently, this approach could become a sentinel pathophysiologic marker identifying preclinical or early changes of AD before the accumulation of beta amyloid or hpTau. It would also provide a simple, inexpensive and noninvasive means to follow patient progress. The obstacles facing fast imaging of this sort includes a susceptibility artifact created by the paramagnetic effect of macromolecules, calcium, etc., within the voxel [57]. This can be compensated in part by incorporating a larger region of interest (ROI) in the analysis. Even so, choice of precise location of data gathering over all seven sequences must be considered with care to reduce excessive sequence to sequence artifacts. Although pulse volume varies intersubject, it is negated by determining the slope of average signal clearance over the seven acquisition times (Figure 3). The rate of clearance is directly comparable across subjects, eliminating qualitative comparisons. Likewise, correcting for cardiac pulse rate is not necessary. Given the rapidity of data collection in a sequence, motion artifact is negligible. Post processing using the elliptical ROI tool is cumbersome with our PACS system, mainly due to the inability to store the generated ROI region for use on sequential sequences. Thus, the ROI had to be carefully redrawn each time [46]. There can be significant variability in signal average within an anatomic site and so avoiding the susceptibility artifact and subarachnoid/ventricular spaces can be challenging as well. Choosing a large ROI dependent on the brain region studied compensates for some of the intravoxel signal variability encountered. The ROI dimensions must be held constant in each brain region for all subjects (Figure 3). That said, with wider adoption of this methodology, simple computer programing fixes would reduce the post processing labor time substantially.

3D ASL MRI provides a novel noninvasive, and inexpensive method to study the earliest pathophysiologic changes in AD. Its full potential will require further development and validation, however. The major limitations, managing the susceptibility artifact and low signal availability. As new therapeutic efforts to reverse the BBB leak arise, the need to screen at risk (preclinical) or AD patients with early disease (be it familial or sporadic) and then follow the results of intervention with simple, safe, economical and reliable testing will be necessary. Cognitive testing alone lacks sufficient sensitivity in preclinical dementia to fulfill that role [60]. Combining it with a sensitive imaging test reflecting physiologic changes would be quite powerful. Further, the utility of identifying vascular leak and altered glymphatic flow opens new avenues into the investigation of other CNS disease processes.

## Figures and Tables

**Figure 1 diagnostics-11-01888-f001:**
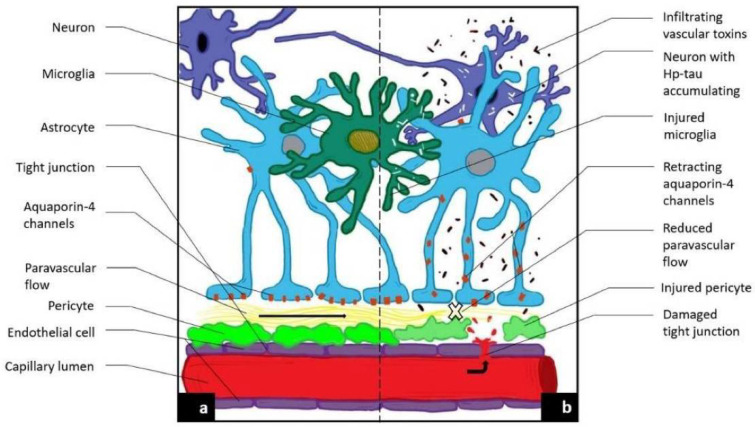
Novel MRI techniques identifying vascular leak and paravascular flow reduction in early Alzheimer disease. (**a**) Demonstrates normal anatomy and physiology. (**b**) Demonstrates the pathological progression to late-state AD including the development of a BBB leak through tight junctions, the retraction of aquqporin-4 channels with reduction in paravascular outflow, and accumulation of Aβ and Hp tau.

**Figure 2 diagnostics-11-01888-f002:**
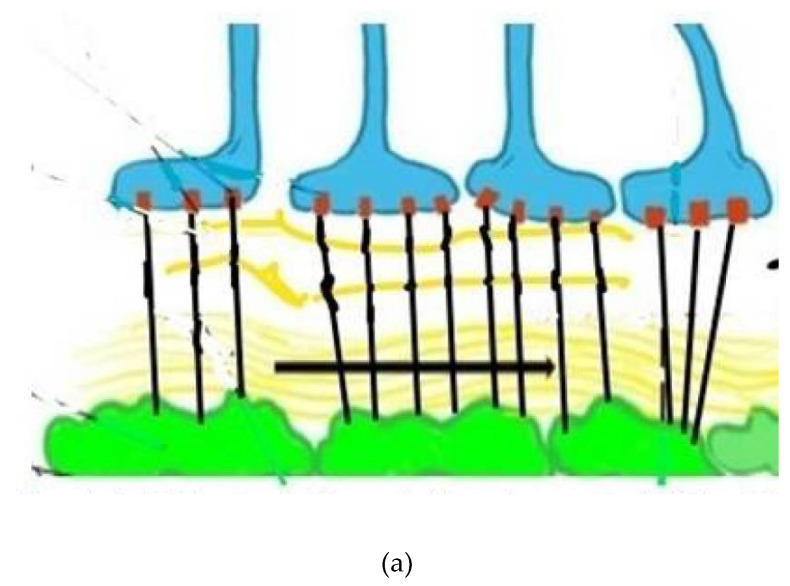
(**a**) Depicts normal aquaporin attachments in the astrocyte end feet before BBB leak and resulting retraction afterwards. Glymphatic channels and anchoring of the aquaporin 4 channels via pericyte protein attachments. The glymphatic space (white and wavy yellow) which exists between the basement membranes of the astrocytes (blue) and pericytes (green), both surrounding capillary and small venous and arterial vessels (not shown). (**b**) Depicts damaged pericytes from vascular inflammation and BBB leak with loss of the tethering proteins and retraction of the AQ4 water channels back into the astrocyte soma with resultant loss of glymphatic flow. (Black lines represent tethering proteins and black dots represent leaked restricted proteins). This sequence of disrupted BBB illustrates the initial pathophysiologic damage leading to the leak and sequestration of DAMPS and PAMPS triggering altered proteomic synthesis and degradation and, ultimately, cognitive dysfunction.

**Figure 3 diagnostics-11-01888-f003:**
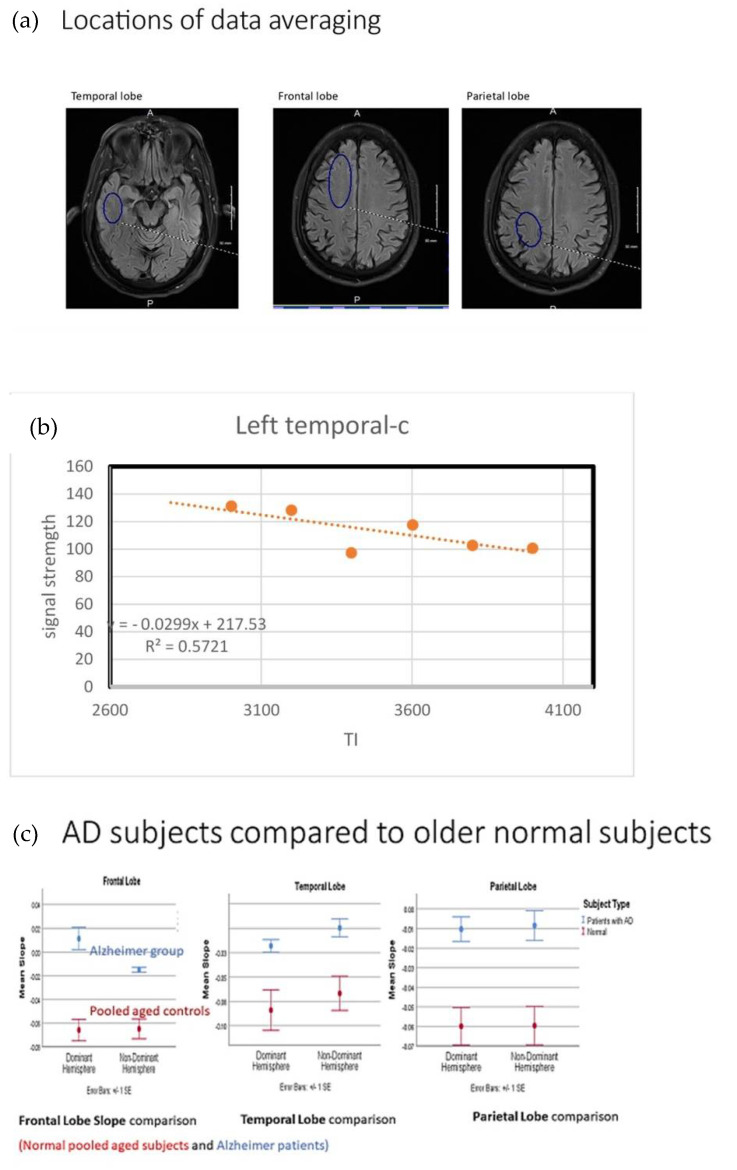
(**a**) Region of interest ROI (blue ovals) size and configuration obtained from the temporal, frontal and parietal lobes. (**b**) Example of ROI plot and determination of slope (rate of signal loss) and correlation coefficient. (**c**) Comparison of rate of signal loss in AD to age matched controls *p* < 0.001 for all but the L temporal (0.067).

**Table 1 diagnostics-11-01888-t001:** Comparison of dynamic contrast imaging and #D TGSE PASL.

MRI Sequence Type	Contrast Agent	Information Sought	Duration of Sequence Acquisition	Duration of Study	Artifact Type	Reproducibility	Cost/Scan
Dynamic Contrast Imaging (DCI)	Exogenous Gadolinium	Presence of BBB leaked contrast K_transfer_ coefficient	16 min per sequence	30+ min for two sequences	Motion artifact, intercompartment contrast equilibrium determination	Yes	High due to need for contrast agent
3D Arterial Spin Labeling (3D ASL)	Endogenous Proton labeling	Delay of labeled proton clearance	2 min per sequence	15 min for seven sequences	Low S/N, susceptibility artifact	Yes	Low

Chart above highlights the pros and cons of the two major techniques for identifying the BBB leak and reduced glymphatic flow rates in preclinical and early AD.

## Data Availability

This review did not include new data. All data from the pilot study with reasonable request is available from the author.

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
