# Peer review of "Utilizing 3D Arterial Spin Labeling to Identify Cerebrovascular Leak and Glymphatic Obstruction in Neurodegenerative Disease"

_diagnostics, 2021, doi:10.3390/diagnostics11101888_

Round 1

Reviewer 1 Report

Paper requires some revisions. See below:

  • In the Introduction section it should be better highlighted the aim of this paper. Please improve.
  • Figure 2. "Glymphatic channels and anchoring of the Aquaporin 4 channels via pericyte attachments" It seems that in this figure it is reported only a pathological progression. If yes, revise.
  • Lines 230-233: "All but the dominant temporal lobe showed difference showed significance to the level p <0.001 when comparing the AD subjects with age matched controls" How do the authors  explain this result?
  • Lines 77-80: "An orderly approach to identifying early sporadic AD includes surveillance of high-risk groups such as those with poorly controlled Diabetes mellitus, hypertension, head injury, morbid obesity, advanced age, ... [27, 28]." But also brain tumor, please consider this important ref:  Clinical Risk and Overall Survival in Patients with Diabetes Mellitus, Hyperglycemia and Glioblastoma Multiforme. A Review of the Current Literature. Int J Environ Res Public Health. 2020 Nov 17;17(22):8501. doi: 10.3390/ijerph17228501.
  • There are no conclusions, please improve. What's new? What does it add new to the literature?
  • Does this paper have some limitations? Please report them.

Author Response

  • Thank you for your insightful and careful review of my paper.
  • In the Introduction section it should be better highlighted the aim of this paper. Please improve.
  • I have made extensive changes to hopefully better clarify the major need this technique addresses for the early treatment of AD aiming at novel pathologic targets.
  • Figure 2. revision
  • I have revised figure 2 so that the normal and pathologic conditions are more clearly defined in both the figure and the caption.
  • Lines 230-233: "All but the dominant temporal lobe showed difference showed significance to the level p <0.001 when comparing the AD subjects with age matched controls" How do the authors  explain this result?
  • I have addressed this explaining that the non-significance may either be by chance , or to an as yet undetermined mechanism.
  • Lines 77-80
  • Thank you for that excellent reference, and I have added it as well as adding in the text glioma as a cause.
  • There are no conclusions, please improve. What's new? What does it add new to the literature?
  • Does this paper have some limitations? Please report them.
  • I have added and revised the conclusions to show the need this ASL procedure fills for early identification and follow up of preclinical AD when future therapeutic trials repairing the BBB leak develop. I have also discussed the limitations of the ASL technique as it exists now. 
  •  

Reviewer 2 Report

Interesting review on an important topic. No major comments/changes from my end, but would need editing for minor grammatical issues. 

Author Response

  • Interesting review on an important topic. No major comments/changes from my end, but would need editing for minor grammatical issues.

Thank you so much for reviewing this article and your kind words.  I have made a careful review and corrected hopefully all of the grammatical errors. 

Round 2

Reviewer 1 Report

Good revision.